# Equine Parvovirus-Hepatitis Screening in Horses and Donkeys with Histopathologic Liver Abnormalities

**DOI:** 10.3390/v13081599

**Published:** 2021-08-12

**Authors:** Verena Zehetner, Jessika-M. V. Cavalleri, Andrea Klang, Martin Hofer, Irina Preining, Ralf Steinborn, Anna S. Ramsauer

**Affiliations:** 1Internal Medicine, University Equine Clinic, University of Veterinary Medicine, 1210 Vienna, Austria; verena.zehetner@vetmeduni.ac.at (V.Z.); irina.preining@vetmeduni.ac.at (I.P.); sophie.ramsauer@vetmeduni.ac.at (A.S.R.); 2Department for Pathobiology, Institute of Pathology, University of Veterinary Medicine, 1210 Vienna, Austria; andrea.klang@vetmeduni.ac.at; 3Genomics Core Facility, VetCore, University of Veterinary Medicine, 1210 Vienna, Austria; Martin.hofer@vetmeduni.ac.at (M.H.); Ralf.steinborn@vetmeduni.ac.at (R.S.)

**Keywords:** EqPV-H, hepatopathy, neoplasia, hepatitis, horse, *ungulate copiparvovirus 6*

## Abstract

There is strong evidence that equine parvovirus-hepatitis (EqPV-H) is associated with the onset of Theiler’s disease, an acute hepatic necrosis, in horses. However, the impact of this virus on other hepatopathies remains unknown. The objective of this retrospective study was to evaluate the prevalence and quantify the viral loads of EqPV-H in formalin-fixed, paraffin-embedded equine and donkey livers with various histopathologic abnormalities. The pathologies included cirrhosis, circulatory disorders of the liver, toxic and metabolic hepatic diseases as well as neoplastic and inflammatory diseases (*n* = 84). Eight normal liver samples were included for comparison as controls. EqPV-H DNA was qualitatively and quantitatively measured by real-time PCR and digital PCR, respectively. The virus was detected in two livers originating from horses diagnosed with abdominal neoplasia and liver metastasis (loads of 5 × 10^3^ and 9.5 × 10^3^ genome equivalents per million cells). The amount of viral nucleic acids measured indicates chronic infection or persistence of EqPV-H, which might have been facilitated by the neoplastic disease. In summary, this study did not provide evidence for EqPV-H being involved in hepatopathies other than Theiler’s disease.

## 1. Introduction

Theiler’s disease, also known as equine serum hepatitis, is an acute and potentially life-threatening fulminant hepatic necrosis. The disease was first reported by Sir Arnold Theiler in 1918 in South Africa, when horses developed severe acute hepatitis and fulminant liver failure after administration of convalescent equine antiserum during vaccination studies to prevent African horse sickness. Affected livers were characterised by severe centrilobular hepatic necrosis [1]. Since then, Theiler’s disease has been reported in many countries worldwide and has been mostly associated with prior administration of equine - origin biologic products such as equine plasma, tetanus and botulism antitoxins, antiserum against Streptococcus equi and allogenic stem cells [1,2,3,4,5,6,7,8]. Nevertheless, there have been reported cases of Theiler’s disease, which did not receive any equine biologic products and in horses that were in contact with affected horses [1,9,10]. These incidences implied that an infectious and transmissible pathogen might be the cause for the development of Theiler’s disease, but a causative agent was not detected for many years [11].

In 2018, Divers et al. detected an unknown virus in serum and liver samples obtained from a horse that died of Theiler’s disease after receiving tetanus antitoxin [8]. The newly recognised virus was named equine parvovirus-hepatitis (EqPV-H). It harbours a small single-stranded DNA genome and is assigned to the species *Ungulate Copiparvovirus 6*, genus *Copiparvoviridae* (*Parvoviridae* family) [8,11]. EqPV-H has been detected in all but one of the recently reported cases of Theiler’s disease and has therefore been linked to the onset of the disease [7,8,9,10,12,13]. In Europe, viral loads of 1.26 × 10^4^ to 2.04 × 10^9^ genome equivalents (GE)/million cells were detected in liver tissue of four horses suffering from Theiler’s disease [13]. In healthy horses in the USA, China, Germany and Austria a DNA prevalence for EqPV-H has been reported between 7% and 17% [8,14,15,16,17,18] compared to a markedly higher DNA prevalence between 54% and 79% in farms in the USA and Canada with recently documented cases of Theiler’s disease [10,12,19]. Current research is focused on disease outbreaks, possible transmission and seroprevalence in horse populations.

In experimentally infected horses a peak viraemia approximately five weeks after infection was observed. The highest viral load was found in serum and liver, which indicates hepatotropism [19]. Lower amounts of viral DNA were also detectable in various other organs and body fluids. As late as 15 weeks after infection, EqPV-H was still detectable in the majority of tissues tested [19]. The onset of hepatitis has been linked to seroconversion and a decrease in viraemia, which further supports the hypothesis of EqPV-H as the causative agent for this disease [19].

All recent findings suggest that EqPV-H is associated with Theiler’s disease. However, there are no studies available so far that were designed to answer the question, whether the hepatotropic EqPV-H might also be associated with other hepatic diseases. Therefore, the aim of this retrospective study was to evaluate the occurrence and to determine the load of EqPV-H in livers of horses and donkeys showing histopathologic abnormalities other than Theiler’s disease.

## 2. Materials and Methods

### 2.1. Sample Collection and Histopathologic Evaluation

The samples archive of the Institute of Pathology was searched for horses and donkeys that were euthanised between 2003 and 2019 at the University Equine Hospital of the Vetmeduni Vienna with various liver pathologies. All available histopathologically abnormal formalin-fixed, paraffin-embedded tissue (FFPE) samples were included in this study. In addition, randomly selected liver samples with normal histopathology served as controls. Initial histopathologic examination was performed following euthanasia of the horse or donkey as part of the histopathological work-up of the patient. To improve standardisation of this study, samples were independently re-evaluated by an experienced pathologist. 

Tissue slides were stained with haematoxylin and eosin and categorised into seven groups based on histopathologic evaluation (group 1: primary and secondary neoplastic diseases, group 2: inflammatory diseases, group 3: cirrhosis, group 4: circulatory disorders of the liver, group 5: toxic and metabolic hepatic diseases, group 6: multiple diseases, and group 7: normal liver tissue). Neoplastic diseases (group 1) included primary neoplasia of the liver or bile ducts as well as secondary neoplasia and tumour-like lesions. The inflammatory diseases (group 2) comprised leucocytic inflammatory cell infiltration of hepatic parenchyma and biliary tract. Cirrhosis (group 3) was characterised by nodular parenchymal regeneration together with fibrosis and consequential disruption of the liver architecture. Circulatory disorders of the liver (group 4) predominantly included cases of subacute and chronic liver congestion. Toxic and metabolic hepatic diseases (group 5) were defined as degenerative processes with reversible and irreversible cytopathology of hepatocytes such as lipidosis, amyloidosis and vacuolation. Samples that were diagnosed with more than one of the stated histopathologic abnormalities were assigned to group 6. Group 7 comprised normal tissue samples.

### 2.2. Detection of EqPV-H by Real-Time PCR

Viral nucleic acid was extracted from 10 × 10 μm paraffin-embedded liver specimens using QIAamp^®^ DNA Micro Kit according to the manufacturer‘s instructions (Qiagen, Hilden, Germany). Extracts were stored at −20 °C until further analysis. Viral load of EqPV-H was related to cell number using *TTC17*, a gene that exists in mammalian species at a single copy per haploid nuclear genome. The single-copy calibrator gene was selected from a catalogue that uses various databases and available genomic data to intuitively determine single-copy orthologs between different phylogenetic groups and species [20]. The EqPV-H assay was designed according to all published EqPV-H complete coding sequence variants from USA, China and Austria to date, and a single-stranded oligonucleotide containing the EqPV-H target sequence was used as positive control. Information regarding the oligonucleotide sequences of the assays and the EqPV-H positive control are provided in Table 1.

Real-time PCR was performed in a volume of 15 μL composed of 1 × PCR buffer B2 (Tris-HCl, (NH_4_)_2_SO_4_ and Tween 20; Solis Biodyne, Tartu, Estonia), 3.5 mM MgCl_2_, 200 µM of each dNTP (Solis Biodyne), 250 nM of each primer and 200 nM hydrolysis probe (Integrated DNA Technologies, Leuven, Belgium), 1 U HOT FIREPol^®^ DNA polymerase (Solis Biodyne) and 3 μL template DNA. Cycling conditions consisted of an initial 15-min incubation at 95 °C for polymerase activation and template denaturation followed by 45 amplification cycles (denaturation: 95 °C for 15 s, annealing/elongation: 60 °C for 60 s). Real-time PCR was run on the qTOWER3/G Real-Time PCR Thermal Cycler operated by the software qPCRsoft384 (v1.2.3.0) (Analytic Jena, Jena, Germany). Experimental samples were measured in triplicate. Equine DNA served as positive control for the *TTC17* assay. Contamination was monitored by a no-template control. A standard deviation threshold of less than 0.4 was adopted for the *Cq* values of the technical triplicates. In case of a higher standard deviation, the replicate that deviated the most from the mean was excluded.

To generate a standard curve for the EqPV-H real-time PCR assay, sequential dilutions of a synthetic oligonucleotide were amplified. The parameters of the resulting regression line facilitated determination of the assay’s amplification efficiency as well as the detection limit and helped to adjust the amount of template DNA for subsequent quantitative measurement based on digital PCR (Figure 1). Furthermore, monitoring based on real-time PCR was performed for additional organs available from positive cases.

### 2.3. Quantification of Cellular Viral Load Using Digital PCR

Digital PCR was performed in Sapphire chips on the Naica Crystal Digital™ PCR System (Stilla Technologies, Villejuif, France). The mastermix contained 1 × PerfeCTa Multiplex qPCR ToughMix (Quanta Bio, Beverly, USA) with 100 nM fluorescein, 800 nM of each primer and 250 nM hydrolysis probe for each target (*TTC17* and EqPH-V) and 9 µL template. Following droplet generation, amplification was carried out according to the following protocol: 95 °C for 10 min, 45 cycles of 95 °C for 10 s and 60 °C for 40 s. Endpoint fluorescence values measured on the Naica Prism 3 reader were analysed by Crystal Miner software (v. 2.4.0.3; Stilla Technologies, Villejuif, France). The software’s standard settings were applied to export the copy numbers for the two targets.

## 3. Results

### 3.1. Clinical Records and Histopathologic Evaluation of Liver Samples

The study population comprised 87 horses and 5 donkeys. Clinical records included age, breed and sex of the animals, clinical diagnosis, activity of plasma liver enzymes and liver-associated blood-biochemical parameters, as well as the primary cause for euthanasia and histopathologic liver findings (Table A1). Clinical diagnosis of liver disease was the cause for euthanasia in 27 cases. In the remaining patients, liver abnormalities were an incidental finding at necropsy and horses were euthanised for unrelated reasons. Nine out of 17 liver samples initially assessed as unremarkable, were found to be histopathologically abnormal in the second review performed during the course of this study.

Based on histopathologic evaluation, the 92 liver samples included in the study were assigned to the following seven groups: neoplastic diseases (*n* = 20, group 1), inflammatory diseases (*n* = 24, group 2), cirrhosis (*n* = 5, group 3), circulatory disorders of the liver (*n* = 4, group 4), toxic and metabolic hepatic diseases (*n* = 14, group 5), multiple diseases (*n* = 17, group 6) and normal liver tissue (*n* = 8, group 7). Samples that were assigned to group 6 were simultaneously diagnosed with at least two histopathologic features of groups 1 to 5 and were predominantly associated with inflammatory disorders (Table A1).

### 3.2. Virus Detection by Real-Time PCR and Copy Number Quantification by Digital PCR

From the study cohort (*n* = 92), two EqPV-H-positive livers were identified by real-time PCR. Both belonged to group 1 comprising neoplastic disease specimens. The samples, hereafter referred to as samples #1 and #2, produced *Cq* values at the upper end of the quantitative dynamic range (29.96 ± 0.20 and 30.08 ± 0.34, respectively; Figure 1). Copy number measurement by digital PCR (Figure 2) determined EqPV-H genome equivalents per million cells of 5 × 10^3^ and 9.5 × 10^3^ (samples #1 and #2, respectively; Table 2). Additional organs available from the two horses included heart, lung, spleen and intestine of case #1 and intestine of case #2. Real-time PCR detected a very low amount of viral DNA in these tissues. In detail, *Cq* values at or beyond the limit of quantification were determined for heart, lung, spleen and intestine of case #1 (32.58 ± 0.46, 33.57 ± 1.22, 32.62 ± 1.15 and 34.62, respectively) and the intestine of case #2 (35.23).

For each of the two intestines, only for one of the three technical replicates a real-time PCR signal was obtained. This indicated that the amount of virus in the sample was at the detection limit of the assay.

### 3.3. Clinical History and Pathological Findings of Horses with EqPV-H Positive Liver Tissue

#### 3.3.1. Case #1

A 21-year-old Warmblood gelding was referred to the University Equine Hospital of the Vetmeduni Vienna due to acute colic and anorexia. The gelding was in good body condition. Clinical examination revealed an elevated heart and respiratory rate (heart rate: 64 beats/min; respiratory rate: 44 breaths/min), mildly elevated rectal temperature (38.1 °C), prolonged capillary refill time and diminished gastrointestinal sounds. Dependent peripheral oedema of the distal extremities and ventral abdominal oedema were present. The following plasma liver enzymes and liver associated blood-biochemical parameters were measured in this case and were within normal limits: glutamate-dehydrogenase, gamma-glutamyltransferase, triglycerides and total protein. Albumin concentration was reduced (14.1 g/L, reference range: 24 to 45 g/L). Diagnostic workup was highly suspicious for abdominal neoplasia and the horse was euthanised. On necropsy severe ascites and a whitish, firm, indistinctly circumscribed mass with the dimensions of approximately 5 cm × 5 cm × 15 cm was found which was located between stomach and spleen as well as multiple, similar, nodular masses, up to 6 mm in diameter. Masses were attached to the serosa of the abdominal cavity and visceral serosa of liver, spleen and intestine, within the liver and spleen parenchyma and the perirenal fat. Histologic examination of these masses revealed well differentiated epithelial tumour cells arranged in tubules with mucus secretion, surrounded by intensive desmoplasia (Figure 3A). In some locations metaplastic ossification of the stromal tissue was also evident. The single-layered, cuboidal to columnar, mucus-secreting tumour cells were bearing hyperchromatic, round or ovoid, basal oriented nuclei (Figure 3B). Mitotic figures were present in low numbers. On histopathology, the masses were diagnosed as ductal adenocarcinoma with stromal osseous metaplasia and transcoelomic peritoneal metastases. The masses most probably derived from the intestinal mucosa.

#### 3.3.2. Case #2

A 19-year-old Warmblood gelding was referred to the University Equine Hospital of the Vetmeduni Vienna for further evaluation of elevated liver enzyme activities and weight loss. On clinical examination the horse was lethargic, in poor body condition, normothermic, normocardic and had a physiologic respiratory rate and effort, reddened mucous membranes, diminished gastrointestinal sounds and a left-sided holodiastolic heart murmur. Measurement of liver specific plasma enzyme activities detected an elevation of gamma-glutamyl transferase (132 U/L, reference range: <30 U/L), glutamate dehydrogenase (20.08 U/L, reference range: <13 U/L) and aspartate-aminotransferase (619 U/L, reference range: <550 U/L).

Transcutaneous abdominal ultrasound revealed hepatomegaly, with the liver being detectable from the 5th to the 13th intercostal space on the left side of the abdomen. Liver tissue had an heterogeneous patchy appearance with hyperechoic areas. Multiple rounded, hyperechogenic, not well-circumscribed and partly vascularised masses of various sizes were detectable within the liver parenchyma (Figure 4). Ultrasonographic findings were highly indicative of liver neoplasia. The owners decided against further diagnostics and opted for euthanasia. Pathological examination detected multiple nodular, firm and whitish-grey, masses of various sizes up to 20 cm in diameter in the liver parenchyma. One white, firm, nodular mass with a diameter of 2 cm was found in the lung parenchyma. Another such mass of 10 cm in diameter was found in the mesocolon. The other organs were unremarkable. Histologic examination of these masses revealed that they were comprised of malignant, mesenchymal cells separated by collagenous stroma (Figure 5A), arranged in a pericapillar and perivascular whorling pattern (Figure 5B). Mitotic figures were not detected within 10 high-power fields and cellular pleomorphism was low. The tumour cells immunohistochemically displayed expression of smooth muscle actin and some of them additionally demonstrated desmin-antigen-reactivity. On the basis of the perivascular pattern and expression of contractile proteins, the tumour was diagnosed as metastasising perivascular wall tumour.

## 4. Discussion

Recently, strong evidence has been presented indicating that EqPV-H is the causative agent for the onset of Theiler’s disease [7,8,9,10,13]. In human medicine, viral infection is one of the most common causes for hepatitis. Reported hepatotropic viruses include hepatitis A, B, C, D and E virus, but also non-hepatotropic viruses such as adenovirus, cytomegalovirus and herpes simplex virus can lead to acute hepatitis and fulminant hepatic failure [21,22]. Depending on the infectious agent and the degree of severity of the disease, histopathologic findings may include signs of acute or chronic hepatitis, liver cirrhosis, congestion and even hepatocellular carcinoma formation [21,22]. As human hepatopathy-associated viruses can lead to widespread manifestations, the determination of the influence of the hepatotropic EqPV-H on other liver diseases in equids beside Theiler’s disease and subclinical hepatitis was the main focus of this investigation. 

Our cohort consisted of 8 normal equine livers and 84 horse/donkey livers with various hepatopathies. Only in 2/92 formalin-fixed paraffin-embedded (FFPE) liver sample EqPV-H DNA was detectable. Both positive cases were diagnosed with abdominal neoplasia and liver metastasis. The general weakness and immunosuppression that accompany neoplastic disease may possibly facilitate secondary infection with EqPV-H. In humans, cancer is associated with a higher susceptibility to opportunistic infections [23], which may also be the case in equine patients. Further studies in humans have demonstrated that the immune response was disrupted with different tumour types and reduced T-cell activation during viral or bacterial infection has been detected in mice [24]. Thus, EqPV-H may not only infect horses suffering from neoplastic disease more easily, but also, persistence of the virus may be supported or even benefit from an insufficient immune response. Further research is warranted in order to test this hypothesis, as only two livers tested positive for EqPV-H in this study and therefore cannot provide sufficient evidence.

In human medicine, a few viruses are associated with the onset of tumours. Regarding the liver, chronic infections with hepatitis B and C viruses can lead to hepatocellular carcinoma [25]. The question as to whether EqPV-H could potentially be the inciting cause for the development of primary liver tumours in equids was not the objective of this study and needs to be evaluated by further research. In our study, both samples that tested positive for EqPV-H were from horses diagnosed with secondary liver neoplasia. Only two primary liver neoplasms were included in group 1, and both of these cases that were diagnosed as adenocarcinoma of the biliary ducts were negative for the presence of EqPV-H. The remaining samples of group 1 were diagnosed with metastases in the liver, while the primary neoplasia originated from lymphoma, hamartoma, carcinoma, round cell neoplasia, leukaemia, hemangiosarcoma, adenocarcinoma, squamous cell carcinoma and spindle cell carcinoma (Table A1).

Case #1 suffered from a ductal adenocarcinoma with metastases in multiple abdominal organs, including the liver parenchyma. As liver enzymes were within normal limits, acute EqPV-H infection was not suspected, and the presence of EqPV-H seemed to be either a co-incidence or a contributing factor of this horse’s disease. Very low amounts of EqPV-H could also be detected in the tissue of the spleen, where tumour metastasis was present, as well as in heart and lung tissue of this horse. This is similar to the findings of Tomlinson et al. [19], in which EqPV-H DNA was detected in several other tissues (lung, spleen, kidney, heart and intestine) in addition to the liver. As in our study, of all the infected tissues, the highest viral load was found in the liver, further supporting the hepatotropism of this virus. Persistence of the virus was suspected by Tomlinson et al., since viral DNA was still detected in serum, liver, lung, spleen, bone marrow and other tissues 15 weeks after experimental infection [19]. In analogy, human parvovirus B19 can persist in serum and tissues for up to years after infection [26,27,28]. Here, we provide further evidence for the persistence of EqPV-H in the liver and other organs, supported by an absence of acute infection in combination with a low concentration of cellular viral DNA. Earlier studies have detected EqPV-H DNA in serum samples originating from clinically normal horses that were in contact with the horses diagnosed with EqPV-H-associated Theiler’s disease [10,12]. Whether the case presented here might have suffered from unrecognised hepatitis prior to hospitalisation cannot be determined retrospectively, however, an acute infection seemed unlikely, as glutamate dehydrogenase and gamma-glutamyl transferase levels were within normal limits in the serum. 

Case #2 was referred due to an elevation in liver specific enzyme activities and ongoing weight loss. Metastasising perivascular wall tumour was diagnosed on histopathology. Smooth muscle cells of the colonic wall were suspected as the origin of the neoplasia. Furthermore, neoplastic cells were noted infiltrating the liver parenchyma. In this case, only liver and colon samples were stored and available for this study, and EqPV-H was only detected in the liver tissue. Increased liver specific enzymes can be explained by the infiltration of liver tissue by neoplastic cells and the resulting damage and cell loss of liver parenchyma. There was no evidence of inflammation on histopathologic evaluation. Thus, persistence of the virus in liver tissue or low-grade chronic infection of the horse seems possible.

Recently, viral loads ranging from 1.26 × 10^4^ GE/million cells to 2.04 × 10^9^ GE/million cells were reported for FFPE liver samples obtained from four horses suffering from Theiler’s disease [13]. The diagnosis was based on clinical signs, serum biochemistry changes and histopathologic findings [13]. In our study, the two liver samples that were clearly positive for EqPV-H nucleic acid had viral loads of 5 × 10^3^ and 9.5 ×10^3^ GE/million cells. The viral loads in these two cases were slightly lower than previously described for acutely infected horses by Vengust et al. [13], which could be another indication of a chronic infection. This is in line with the study of Tomlinson et al. reporting viral loads of 1 × 10^4^ to 1 × 10^5^ in liver tissue 5 weeks after experimental infection [19]. Viraemia developed within a median time period of 2.3 weeks after inoculation of the virus. After 6.5 weeks, eight out of ten horses developed hepatitis, which was defined as two liver enzymes having values above the reference range [19]. One horse also suffered from clinical signs including icterus, mild lethargy and inappetence [19]. After 15 weeks, viral loads of 1 × 10^1^ to 1 × 10^5^ GE/million cells were still detectable in the liver which again highlights the possibility that EqPV-H infection can become chronic with persistence of the virus [19], as was assumed in our two cases, which exhibited low amounts of the virus.

A limitation of this study was the storage of tissue until analysis, which might have had an influence on measured viral loads. As the study was performed retrospectively, all tissue samples were stored embedded in paraffin between 1 and 17 years at room temperature, which could have affected the amount of detectable virus. The two positive samples derived from horses euthanised in the year 2018 had been stored for two years until analysis. Even though storage might have impaired the amount of viral DNA, EqPV-H was successfully detected by the newly designed RT-PCR assay in two livers.

Another interesting fact is that both positive cases were presented to the University Equine Hospital of the Vetmeduni Vienna in the year 2018 with only one month apart from each other in February and March, respectively. This might only be a coincidental finding. An infection during hospitalisation seems unlikely as the incubation period is approximately 4 to 13 weeks, and both patients were euthanised soon after admission (case #1: after 3 days, case #2: at the day of admission). Seasonal occurrence of EqPV-H-associated Theiler’s disease during late summer to early fall months has been reported [10,12,13], but the exact time and season of infection in the presented cases cannot be determined retrospectively. Badenhorst et al. [15] reported an EqPV-H DNA prevalence of 8.9% in sera of healthy Austrian horses which is similar to the results of other surveillance studies [8,14,15,16,17,18]. Therefore, a considerably higher risk of infection in Austrian horses seems unlikely. 

## 5. Summary

Although this study is limited by the number of positive samples and the retrospective study design, the results did not find an association between EqPV-H and other liver diseases apart from Theiler’s disease, as in 82/84 liver samples with histopathologic abnormalities, no EqPV-H viral DNA was detectable. Nevertheless, the fact that both positive samples were detected in the group of 20 horses with neoplastic disease is interesting and future investigations of a potential association or even interaction between EqPV-H infection and neoplastic disease may be warranted.

## Figures and Tables

**Figure 1 viruses-13-01599-f001:**
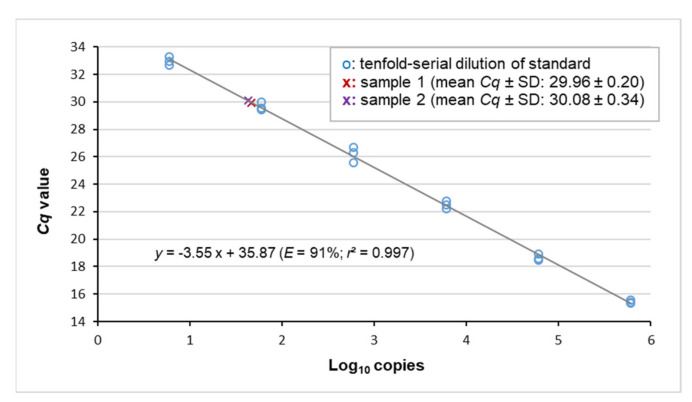
Viral loads of samples #1 and #2 matched the quantitative dynamic range of the assay. The standard curve was obtained by plotting the *Cq* values of serial standard dilutions against the decadic logarithm of the respective copy-number concentration. r: Pearson’s correlation coefficient.

**Figure 2 viruses-13-01599-f002:**
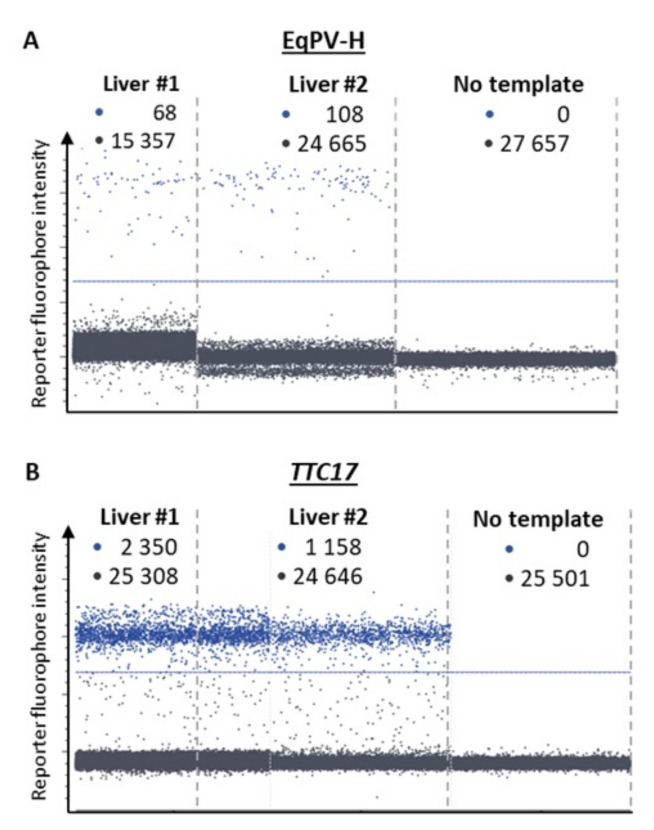
Detection of EqPV-H (**A**) and the cellular calibrator gene *TTC17* (**B**) by dPCR in liver samples #1 and #2. Each panel represents a separate experimental chamber on a chip where the reaction mix containing DNA from liver samples (1:20 dilution for *TTC17*) was segregated into individual droplets and assessed for the presence of EqPV-H and *TTC17*. •: positive samples; •: negative signals, respectively. The blue line designates the arbitrary fluorescence threshold separating the signals from background.

**Figure 3 viruses-13-01599-f003:**
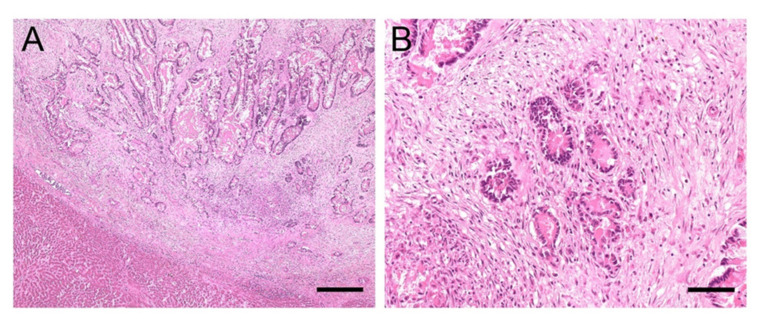
(**A**,**B**). Histology of liver sample #1: Adenocarcinoma metastases. (**A**): Sharply demarcated infiltration of epithelial tumour cells arranged in tubules surrounded by intensive desmoplasia. Mucus secretion can be seen within the tubules; bar = 400 µm. (**B**): Single-layered tumour cells containing round or ovoid, basal oriented nuclei (**B**); bar = 80 µm.

**Figure 4 viruses-13-01599-f004:**
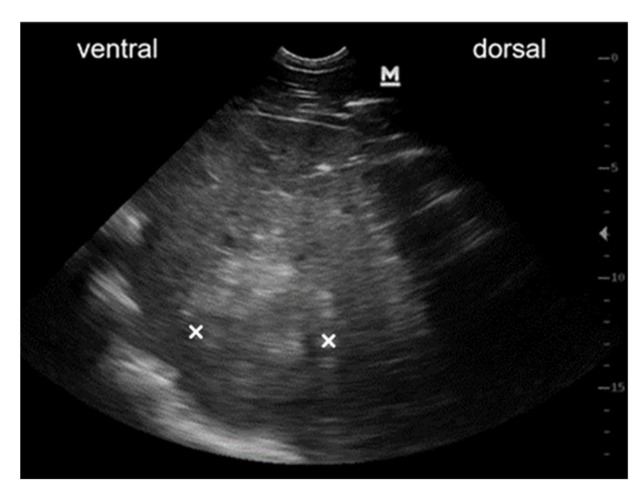
Ultrasonographic image of mass within liver parenchyma. Ultrasonographic image obtained from the right 14th intercostal space of the abdomen at the costochondral border. Liver tissue is highly inhomogeneous with a rounded, hyperechoic and weakly circumscribed mass within the liver parenchyma (×: dorso-ventral expansion of the mass).

**Figure 5 viruses-13-01599-f005:**
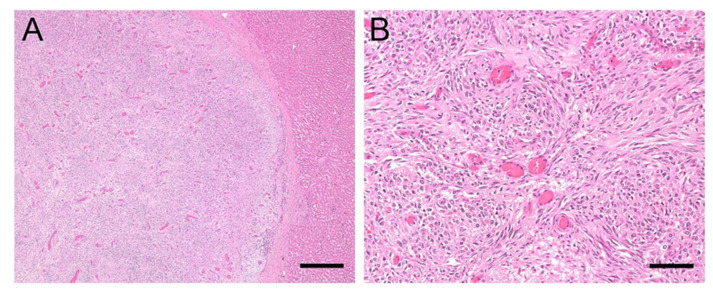
(**A**,**B**). Histology of liver sample #2: Perivascular wall tumour. (**A**): The masses found in the liver were comprised of malignant, mesenchymal spindle cells, separated by collagenous stroma; bar = 400 µm. (**B**): The tumour cells are arranged in a pericapillar and perivascular whorling pattern, bar = 80 µm.

**Table 1 viruses-13-01599-t001:** Detection and quantification assays (real-time PCR and digital PCR) for measurement of cellular EqPV-H load.

Assay	GenBank Identity	5′ to 3′ Sequence of Oligonucleotide	Amplicon Size (bp)
EqPV-H *	NC_040652.1, MG136722.1, MH500787.1 to MH500792.1, MN218583.1 to MN218592.1, MW256660.1 to MW256663.1	F: AAG ATA TGC CGC CAT TTG AA	77
		R: CTG AAA AGG CAT TCC GTC AG	
		P: FAM-CAG AGA AAT /ZEN/ CCT GAG CGG TGG CCT-IBFQ	
		PC: ATC TTC TAT AAA GAT ATG CCG CCA TTT GAA AAG GCC ACC GCT CAG GAT TTC TCT GAC TAT TAT GTT TCT GAC GGA ATG CCT TTT CAG ACT TTG TAT G	
*TTC17*	XM_023653901.1	F: CTG GAC AAC AGC CAT GAC AAA	147
		R: AAG TCT AAG GGC ATC TGA GTC CC	
		P: FAM-CAC AGG GTC /ZEN/ CTC CTC TGC TCC TGTC-IBFQ	

F: forward primer, R: reverse primer, P: probe, PC: positive control (amplified region is under-lined) * Designed against all publicly available EqPV-H variants FAM: 6-fluorescein amidite, ZEN: internal quencher inserted after 9th base of the ZEN™ Dual-Quenched Probe; IBFQ: Iowa Black™ Fluorescent Quencher. Synthesis of oligonucleotides: Integrated DNA Technologies, Leuven, Belgium.

**Table 2 viruses-13-01599-t002:** Digital PCR determination of cellular viral load.

Sample	TTC17 Copies/μL (1:20 Dilution)	EqPV-H Copies/μL	Cellular Viral Load	
			Virus/Cells	GE/10^6^ Cells
**#1**	151.5 ± 6.1	7.54 ± 1.81	1/201	5.0 × 10^3^
**#2**	78.4 ± 4.7	7.46 ± 1.42	1/105	9.5 × 10^3^

## Data Availability

Data is contained within the article.

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
