# Peer review of "Equine Parvovirus-Hepatitis Screening in Horses and Donkeys with Histopathologic Liver Abnormalities"

_viruses, 2021, doi:10.3390/v13081599_

Round 1
Reviewer 1 Report
This manuscript addresses an important question regarding any association between the presence of EqHV-P and liver pathology in horses and donkeys. The study design is robust, using a reasonable number of very well characterised clinical cases and importantly, includes control samples from normal equine livers. Experiments are also well controlled with appropriate positive and negative controls and number of replicates stated. The data are presented clearly and accurately and the conclusions are supported by the data. It was a pleasure to review, thank you.
I would recommend that the following minor corrections are completed prior to publication:
Essential corrections:
Table 2: final column - please amend 106 cells to 106 cells
L339: it may be worth mentioning that the 2 positive cases had been stored since 2018 ie 2 years?
Table S: please replace Islandic and Icelandic horses (cases 43, 56, 60-22 etc)
Optional amendments:
L226 - is inhomogeneous the correct word here? Would heterogeneous be better?
Table 2: TTC17 column - please put the "1:20" dilution on the second line.
Author Response
Dear Reviewer,
Thank you for the review of our manuscript and your recommendations. We appreciate your comments on our work and your valuable input.
We corrected the manuscript due to your suggestions.
Table 2: final column - please amend 106 cells to 106 cells
- “106 cells“ was changed to “106 cells“
L339: it may be worth mentioning that the 2 positive cases had been stored since 2018 ie 2 years?
- The following sentence was added to the manuscript according to your suggestion:
- L337: The two positive samples derived from horses euthanised in the year 2018 and had been stored for two years until analysis.
Table S: please replace Islandic and Icelandic horses (cases 43, 56, 60-22 etc)
- We apoglogize for the spelling mistakes, they were corrected to Icelandic horses.
L226 - is inhomogeneous the correct word here? Would heterogeneous be better?
- Due to your suggestions, the wordings has been changed to heterogeneous.
Table 2: TTC17 column - please put the "1:20" dilution on the second line.
- Table 2 was corrected as you suggested.
Reviewer 2 Report
Line 45: Reference should be (8) not (12)
Line 190: Insert "gastrointestinal for "gut"
Line 191: Was "ventral oedema" notices as "ventral thoracic oedema" or "ventral abdominal oedema" or if both "ventral oedema" is acceptable
Line 194-195: Use S.I. units eg. 1.41 g/dL = (14.1 g/L , reference range : 24-45 g/L)
Line 196: Replace "forearm-sized" with the actual dimensions ? cm x ? cm x ? cm
Line 204: Insert hyphen "mucus-secreting"
Line 206-208: Rewrite sentence: "On histopathology, the masses were diagnosed as ductal adenocarcinoma with stromal osseous metaplasia and transcoelomic peritoneal metastases. The masses most probably derived from the intestinal mucosa"
Line 219: Replace "gut" with "gastrointestinal"
Line 219: Insert hyphen "left-sided"
Line 220-222: Change order of enzymes in sentence to gamma-gutamyl transferase first followed by glutamate dehydrogenase and then aspartate amino transferase. Reflects the order of the more liver-specific enzymes.
Line 232: Prefer "firm white" to ""white firm"
Line 266: Insert "possibly" so reads "may possibly facilitate"
Line 281: Insert "neoplasms" for "neoplasias"
Line 287: Insert "metastases" for "manifestations"
Author Response
Dear Reviewer,
Thank you for the review of our manuscript and your recommendations. We appreciate your valuable input.
We corrected the manuscript due to your suggestions.
Line 45: Reference should be (8) not (12)
- We apologize for the mistake in the reference and changed it to the correct one (8)-
Line 190: Insert "gastrointestinal for "gut"
- The wording was changed according to your suggestions.
Line 191: Was "ventral oedema" notices as "ventral thoracic oedema" or "ventral abdominal oedema" or if both "ventral oedema" is acceptable
- The oedema was present at the ventral abdomen, the wording was changed to „ventral abdominal oedema“.
Line 194-195: Use S.I. units eg. 1.41 g/dL = (14.1 g/L , reference range : 24-45 g/L)
- The units were changed to S.I. units g/L.
Line 196: Replace "forearm-sized" with the actual dimensions ? cm x ? cm x ? cm
- Unfortunatly the exact dimensions of the mass were not measured, therefore the wording was changed to “..mass with the dimensions of approximately 5 cm × 5 cm ×15 cm..“.
Line 204: Insert hyphen "mucus-secreting"
- A hyphen was added to "mucus-secreting"
Line 206-208: Rewrite sentence: "On histopathology, the masses were diagnosed as ductal adenocarcinoma with stromal osseous metaplasia and transcoelomic peritoneal metastases. The masses most probably derived from the intestinal mucosa"
- The sentence in Line 206 – 208 has been changed as you suggested.
Line 219: Replace "gut" with "gastrointestinal"
- The wording was changed according to your suggestions.
Line 219: Insert hyphen "left-sided"
- A hyphen was added to "left-sided".
Line 220-222: Change order of enzymes in sentence to gamma-gutamyl transferase first followed by glutamate dehydrogenase and then aspartate amino transferase. Reflects the order of the more liver-specific enzymes.
- The enzymes have been reordered in the sentence due to your suggestions.
Line 232: Prefer "firm white" to ""white firm"
- The wording has been changed to “firm and whitish-grey“.
Line 266: Insert "possibly" so reads "may possibly facilitate"
- The sentence was changed to …“disease may possibly facilitate secondary..“.
Line 281: Insert "neoplasms" for "neoplasias"
- The wordings was changed to “neoplasms“.
Line 287: Insert "metastases" for "manifestations"
- The wordings was changed to “metastases“.